# Non-Timber Forest Product Livelihood-Focused Interventions in Support of Mangrove Restoration: A Call to Action

**Adolphe O. Debrot** [1,2,*], **Ab Veldhuizen** [3], **Sander W. K. van den Burg** [4], **Charlotte J. Klapwijk** [5], **Md. Nazrul Islam** [6], **Md. Iftakharul Alam** [7], **Md. Nazmul Ahsan** [8], **Moin U. Ahmed** [9], **Selim R. Hasan** [9], **Ratnawaty Fadilah** [10], **Yus R. Noor** [11], **Rudhi Pribadi** [12], **Sri Rejeki** [13], **Ekaningrum Damastuti** [14], **Esther Koopmanschap** [15], **Stijn Reinhard** [4], **Catharien Terwisscha van Scheltinga** [3], **Charlotte Verburg** [16] and **Marnix Poelman** [1]

1   Wageningen Marine Research, Wageningen University and Research, P.O. Box 57, 1780AB Den Helder, The Netherlands; marnix.poelman@wur.nl
2   Marine Animal Ecology Group, Wageningen University and Research, P.O. Box 338, 6700AH Wageningen, The Netherlands
3   Wageningen Environmental Research, P.O. Box 47, 6700AA Wageningen, The Netherlands; ab.veldhuizen@wur.nl (A.V.); catharien.terwisscha@wur.nl (C.T.v.S.)
4   Wageningen Economic Research, P.O. Box 29703, 2502LS The Hague, The Netherlands; sander.vandenburg@wur.nl (S.W.K.v.d.B.); stijn.reinhard@wur.nl (S.R.)
5   Agrosystems Research, Wageningen University and Research, P.O. Box 16, 6700AA Wageningen, The Netherlands; lotte.klapwijk@wur.nl
6   Forestry and Wood Technology Discipline, Khulna University, Khulna 9208, Bangladesh; nazrul17@yahoo.com
7   Aquaculture and Fisheries Group, Wageningen University and Research P.O. Box 338, 6700AH Wageningen, The Netherlands; iftakharku97@yahoo.com
8   Fisheries and Marine Resource Technology Discipline, Khulna University, Khulna 9208, Bangladesh; nazmul_ku@yahoo.com
9   Solidaridad Network Asia, Dhaka 1209, Bangladesh; moin@solidaridadnetwork.org (M.U.A.); selimr@solidaridadnetwork.org (S.R.H.)
10  Blue Forests, Jl. Pengayoman Komplek Mawar Blok A No. 17-18 Panakukkang, Makassar 90222, Indonesia; ratna@blue-forests.org
11  Yayasan Lahan Basah, Wetlands International Indonesia, Jl. Bango 11, Bogor 16161, Indonesia; yus.noor@gmail.com
12  Marine Science Department, Faculty of Fisheries and Marine Sciences, Diponegoro University, Jawa Tengah 50275, Indonesia; rudhi_pribadi@yahoo.co.uk
13  Aquaculture Department, Faculty of Fisheries and Marine Sciences, Diponegoro University, Jawa Tengah 50275, Indonesia; sri_rejeki7356@yahoo.co.uk
14  Environmental Systems Analysis Group, Wageningen University and Research, P.O. Box 47, 6700AA Wageningen, The Netherlands; ekaningrum.damastuti@wur.nl
15  Wageningen Centre for Development Innovation, Wageningen University and Research, P.O. Box 88, 6700AB Wageningen, The Netherlands; esther.koopmanschap@wur.nl
16  Wageningen Livestock Research, Wageningen University and Research P.O. Box 338, 6700AH Wageningen, The Netherlands; charlotte.verburg@wur.nl
*   Correspondence: dolfi.debrot@wur.nl; Tel.: +31-0317-487395

**Abstract:** Mangroves of tropical and subtropical shores and deltas contribute to ecosystem functioning and human wellbeing in numerous ways but continue to be lost or degraded worldwide at a rapid pace. Overexploitation driven by poverty is often the root cause of mangrove destruction and degradation. The negative feedback cycle between destruction and poverty can only be broken by justly valuing current or introducing new sustainable livelihood options to foster long-lasting local support for

mangroves. The large array of non-timber forest products (NTFPs) that mangroves offer have rarely been developed beyond the subsistence level and remain undervalued as "products of the poor". In light of the global trends towards sustainability and bio-economy, today they represent a major business opportunity for forest communities to produce high value-added end-user products. Even though mangrove NTFPs have been recognized to have high potential toward inclusive development and poverty alleviation and to be highly gender-equal, the development of mangrove NTFPs has continued to attract very little funding or research interest. Several ecological characteristics make commercialization of mangrove NTFPs particularly challenging. Production at economies of scale, including quality standards, as well as marketing and value chain management are all essential in order to develop these products beyond their subsistence role. To be most effective, a systems perspective on NTFP development is needed, whereby product-market development occurs in unison and based on a participative, inclusive and fair development approach. The species/product of choice for value-added product-market development in any specific community or area will depend on several factors. To address many of the typical constraints and maximize the chances of success, we suggest that the use of village or district-level cooperatives may be particularly useful. A better use of the untapped potential of mangroves for local livelihoods may form a most convincing advocate for local protection and restoration of mangrove forests. Therefore, funding agencies, governments and researchers alike are called to invest in mangrove NTFP development as a way to locally incentivize sustainable mangrove protection and restoration.

**Keywords:** mangrove NTFP; NWFP; alternative livelihood; ecological constraints; scale mismatch; systems approach

---

## 1. Introduction

Mangroves are a diverse set of uniquely salt- and flooding-tolerant plants forming coastal forests worldwide along sheltered tropical and subtropical muddy shorelines. They grow where no other trees can grow and contribute importantly to total forestation in many countries along rivers and shores [1]. They have long been unsustainably exploited for their wood and used as timber, for fuel and charcoal and for their tannins [2,3]. However, mangroves contribute to human wellbeing in countless other ways such as in terms of protection against natural disasters, land reclamation, livelihoods, healthy nutrition, medicines and access to safe drinking water, as well as cultural and heritage values [4,5]. Mangrove protection and restoration in coastal landscapes interconnects with multiple sustainable development goals (SDGs) related to: sustainable agriculture and food security; sustainable economic growth; response to climate change; conservation, protection and restoration of ecosystems and biodiversity; and partnerships and cooperation for research, and innovation for sustainable use of land and water resources. However, despite the (relatively) widespread awareness of the value of mangroves, more than 25% of original global coverage had already been lost by 2005 [6] and more than 35% of the then still present mangroves having been lost worldwide during the 1980s and 1990s alone [5]. The highest proportions of losses have occurred in Southeast Asia, where 50% of the mangrove quadrants studied showed partial mangrove loss and the key driver of loss was land use for agriculture, aquaculture and urban purposes [7]. The drivers of mangrove loss are quite context-specific and differ from region to region [8]. For instance, for the year 2012, overall land-use cover changes showed that 23% of deforested mangroves were converted to agriculture, 6% to aquaculture, and less than 1% to infrastructure, while 16% were impacted but not converted for any other specific land use and land-use changes were unobservable for the rest of the surface area considered [8]. Findings by Ilman et al. [9] predict that aquaculture (followed by palm oil plantations) will continue as the main driver of mangrove destruction in Indonesia for the next two decades. They stress that this is principally caused by the low productivity of abandoned ponds wherefore farmers

are forced to continue to clear more mangroves for shrimp pond construction. Hence, failure to transform current short-term aquaculture practices into long-term sustainable production or develop mangrove-compatible alternative livelihoods will result in continued loss of mangroves worldwide, with dire consequences for coastal safety and other ecosystem services.

In response to the multifarious negative consequences of mangrove loss, there has been a major drive to restore protective mangrove greenbelts. Recently, several nations have adopted legislation to protect coastal mangroves (e.g., Thailand; [10]), while others have presented ambitious strategic plans for mangrove restoration (e.g., Bangladesh; [11]).

Despite the best intentions, reviews have shown that most restoration efforts remain beset by problems or have outrightly often failed [12,13]. Even so, in the two recent decades, from 1996 to 2016 the rate of global net mangrove loss appears to have amounted to a combined loss of only 1.8% [14], allowing us to conclude that the combined effects of mangrove conservation, restoration and natural recovery must have contributed to these relatively lower recent cumulative global losses. Until a decade ago, the most common purposes of mangrove restoration had been silviculture (for timber and wood) and secondly coastal protection [15]. However, after the massive Indian Ocean tsunami that killed more than 270,000 people in 14 countries in 2004, the main purpose shifted towards coastal protection [16]. The nature-based restoration of livelihoods based on the rich forest resources provided by mangroves has remained a much less important purpose for restoration.

A key issue that has often been overlooked in nature-based, ecologically oriented mangrove restoration is the need for sustainable sociocultural economic integration of mangrove conservation into the livelihoods of local inhabitants [5]. In much of the world's tropics, overexploitation driven by poverty is typically the root cause of mangrove removal and degradation. This poverty of the rural mangrove communities often results from various forms of environmental injustice, including land appropriation by powerful private interests and non-inclusive management approaches [17–20]. Additionally, mangrove restoration often takes place in coastal communities that have suffered economic setbacks either due to environmental exhaustion or natural hazards such as tsunamis or cyclones. Under such conditions, local livelihood options are so reduced that, due to lack of viable alternatives, the local population is forced to revert back to the unsustainable practices that caused environmental depletion and vulnerability to hazards in the first place (Figure 1). Only by interrupting this poverty cycle and introducing sustainable livelihood options can the negative feedback cycle be broken to allow lasting local support for nature-based restoration [21].

Another important complicating factor is the fact that many of the most important economic benefits of mangrove ecosystems (mangrove forests, including their integral component habitats such as creeks, mudflats and lagoons, e.g., [22]) are realized at a distance away from the mangroves proper and accrue to parties not necessarily intimately invested in mangroves. For example, the nursery and productive value of mangroves to fisheries has often been demonstrated and discussed [23–27], but this value is largely exported by tides and fish migration to fisheries located away from the mangroves. Likewise, the value of mangroves in protecting coastal infrastructure and resources typically extends land-inward away from the actual mangrove stands [28–30]. Hence, there often exists a major spatial and sectoral mismatch between those invested in mangrove restoration as a prime example of a nature-based approach and those deriving the bulk of benefits. Such lack of alignment or "scale mismatch" between ecologically-connected systems and management boundaries creates major social and ecological complexities for management [31,32]. This challenge of proper distribution of accrued benefits resulting from nature-based mangrove management has spurned various community-based and benefit-sharing approaches (e.g., community-based forest management, bio-rights) as means to ensure that mangrove management benefits and costs are more equitably spread amongst all stakeholders [13,33–36]. Likewise, the Forest Department of Bangladesh (FD) [37] highlighted the need to share the benefits of forest wild harvest among the communities living around the Sundarbans mangrove forest.

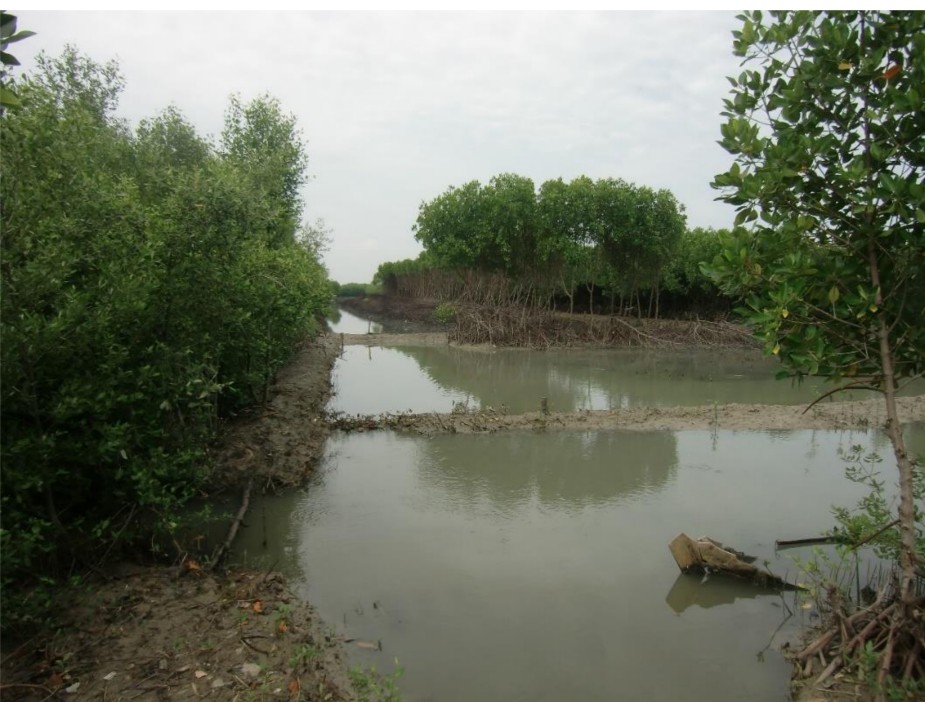

**Figure 1.** Because of the lack of alternative livelihood options, previously restored mangroves are cleared for pond construction in a restoration area on Java, Indonesia. (Photo credit: A.O.D.).

The development of more site-bound, spatially explicit economic benefits of mangrove ecosystems, among which are non-timber forest products (NTFPs), could also significantly help provide local incentives to maintain and protect mangrove systems and to help resolve the problems of scale mismatch. One form of site-bound benefits of mangroves that has recently been receiving much attention is their positive effect on shrimp pond-culture yields. [38–43]. However, very little work has so far been dedicated to developing mangrove NTFPs as viable livelihood options, and the focus has largely been on timber while other forest economic values and other services have been ignored [44–46]. Most mangrove products are almost exclusively used locally for subsistence due to lack of market development and promotion [47]. In several areas of the tropics, however, honey collection or beekeeping is one of the rare areas for which product development has come under recent attention [48–52]. While calculations show that the net economic value of mangroves typically greatly exceeds the value of commercial aquaculture [53], very few sustainable livelihood alternatives have been developed for use by coastal communities, and scalable commercialization of mangrove products is dearly needed for them to be able to truly serve as viable livelihood options [54]. Many authors point to the lack of effort, progress and support in developing NTFPs in general [55–57], while others indicate the lack of integration of sustainable livelihood options in nature or ecosystem-based mangrove conservation and restoration efforts [47,58,59]. Finally, the most recent review on the status of mangroves worldwide shares the perspective that integrating human livelihood needs into mangrove ecosystem conservation is called for to achieve long-term sustainability for mangrove forests [5].

The purpose of this position paper is to present an evidence-based plea to recognize the value of mangrove NTFPs, calling for increasing attention, funding and action towards NTFP development. We develop our argument by (a) illustrating the current value and potential of mangrove NTFPs based on documented mangrove use options for Southeast Asia; (b) assessing the current state of NTFP research and development (worldwide); and (c) discussing ecological properties of mangroves that pose important challenges to NTFP development, thereby also highlighting the need for a systems approach to developing mangrove NTFPs. We conclude with a unified call for increased attention, funding and action in support of mangrove NTFP development.

## 2. Materials and Methods

We compiled a new, more comprehensive and detailed listing of non-timber, plant-based mangrove uses than heretofore available for Southeast Asia and the Pacific by integrating species lists and uses from a number of key literature sources, and we use this as a basis for our discussion of the economic potential represented by NTFPs. This overview was restricted to Southeast Asia but included mangrove-associated species that are part of the mangrove forest flora even if they are not considered strict mangrove species. We did exclude firewood and charcoal from our definition of NTFPs as these are most typically timber-derived products (see [60] for a notable exception). To assess recent developments in mangrove NTFP research (formerly labeled "minor forest products", more recently also labeled "non-wood forest products" or "NWFPs"), we further made a limited (yet illustrative) bibliometric evaluation of research trends (as of 1980) using an online search in English of the Scopus bibliographic database. The research fields examined were as follows: mangroves (all subspecialties), general NTFP (including the following variations: NWFP; nonwood forest products; non wood forest products; nontimber forest products; non timber forest products), mangrove-NTFP, mangrove + fisheries, and mangrove + restoration (incl. rehabilitation). We further limited the search of terms to the title, the abstract and the keywords of an article, so as to exclude papers that only mentioned our targeted research areas solely in passing. Following Roe et al. [61], the exact search terms and Boolean operators were separately documented. The total numbers of published papers classified per research area for the period 1980–2019, on which our graphs were based, were as follows: 20,742 mangrove papers and 2995 NTFP papers. From within the total number of papers addressing the topic of mangroves, we parsed out the following additional subsets: mangrove + NTFP papers, mangrove + fisheries papers, and mangrove + restoration papers. This was clearly not an exhaustive search but was large enough to accurately signal recent research trends. The literature thus traced was reviewed to identify key sources on which to base our discussion of recent concepts applicable to the future of mangrove NTFP research and development. Due to the clear paucity of more in-depth mangrove NTFP research, we borrowed significantly from the available more general NTFP literature (located via additional Google search) to develop our views for this paper.

## 3. Role of Mangrove NTFPs and Plurality of Potential

The economic significance of NTFPs is vast and far-reaching, and a recent reassessment concludes that thanks to the increasing trend towards bio-economy, they represent a major new business opportunity [45]. Worldwide, 1.5 billion people use or trade non-timber forest products [55]. This includes a multitude of plant and animal products derived from mangrove forests. Allowing sustainable harvest of mangrove NTFPs plays an important role in sustainable forest management and conservation strategies [62]. Schreckenberg et al. [63] identified three generally important functions of NTFPs in the context of poverty reduction. First, they serve as safety nets to prevent households falling into greater poverty; second, they help fill income gaps by providing supplemental income; and third, they may even serve as "stepping stones" by pulling people out of poverty when the NTFPs are well-integrated into the local cash economy. For instance, a worldwide survey of 8000 households in developing countries found that reliance on forest subsistence was highest among the 40% lowest income households [64]. In Indonesia there are more than 25,000 villages located in and around forested areas, and 71% of the population of these villages depends on forest resources for their livelihood [65]. Furthermore, it was found that the mean forest and wild product-related income of people living in such forest-rich rural areas can exceed the mean agricultural income [66]. For Myanmar, it was found that poor households were able to obtain up to 36% of their income from community-managed mangrove forests and that when managed inclusively, community-based mangrove forest management has the potential to effectively reduce poverty [67]. Based on a survey of households living in the vicinity of the Sundarbans in Satkhira district, Bangladesh, Rahman et al. [68] found that about 80% of the households depended on mangroves for all or part of their incomes, while nearly 35% of households depended entirely on mangroves. In most cases, as pointed out by Banjade and Paudel [69],

the poor may have better access to NTFPs than to timber because NTFPs can usually be collected free of permits or payments in most community forest settings. Chechina et al. [70] and FD [37] provide examples in which livelihood access to forest resources in the form of collection of NTFPs improved socioeconomic condition whilst also supporting ecological integrity and ecosystem services.

Mangrove NTFPs offer opportunities for livelihood diversification, which is an important adaptation strategy used elsewhere in rural areas [71]. The majority of people living in or near mangrove areas derive their principal income from fishing and related activities, whereas fuelwood and construction materials comprise the next two most widespread uses of mangroves [72]. Despite recent availability of electricity and natural gas, remote coastal communities in many parts of the tropics still depend heavily on mangroves for domestic fuelwood consumption. Similarly, despite the widespread adoption of tin as a lasting roofing material, many coastal communities in Southeast Asia still use fronds from the mangrove palm *Nypa fruticans* for use in roofing and as thatch in walls and floor mats [72,73]. In addition to fuelwood and construction materials, mangroves are also widely valued for their bark (used in tanning and dying), their fiber (to make rayon and paper), and as sources of animal fodder, vegetable foods, and diverse traditional medicines and toxicants [74,75]. Being seasonal in nature, the harvest of these and other mangrove NTFPs is rarely a full-time occupation for the community but a great many rely on these products to meet subsistence needs, while providing others an important income supplement. For instance, Rönnbäck et al. [15] found a high degree of mangrove dependence in coastal communities in Kenya and identified some 24 ecosystem goods derived from mangroves among which foods and medicines ranked the highest. Similarly, in Gambia, 80% of the peri-urban population largely depends on mangroves for a variety of natural products [76]. In the Ayeyarwaddy region of Myanmar, 43% of total household income is generated through selling mangrove forest products [77], whereas almost all households surveyed reported using at least one mangrove community forest product [67]. Such subsistence products included fuelwood, wooden poles, *Nypa* materials, vegetables, fish, crabs, medicinal plants, honey and beeswax [56,67,75]. For villages in Khulna, Bangladesh, mangrove forest income represented 24%, 48% and 74% of total household incomes respectively, for upper, middle and lower-income village households [78]. Around 30% of people in the Sundarbans landscape zone of Bangladesh depend on mangrove NTFPs [37]. Likewise, Datta [58] discusses a wide variety of products harvested by the inhabitants of the Sundarbans in India. Cost–benefit analyses by Chow [60] show that based on direct use values of non-main wood and leaf litter extraction alone (not counting all the other extractive or indirect services), mangrove plantations in Bangladesh are economically justified. Islam et al. [73] show how a single forest product that is harvested only from October to March may provide 28% of annual income to Sundarbans households. Moreover, Olaleye and Omokhua [79] reported the potential of NTFPs in favor of women's empowerment through sustainable use of mangrove resources. Some NTFPs can even be developed by children as reported by FD [37]. Finally, mangrove NTFPs are recognized as being of great potential not only for poverty alleviation but also for inclusive development as they are very gender-equal [17,19,63,80,81].

Examples of non-wood uses of mangrove plants are plentiful (Figure 2). For instance, Kusmana [47] documented more than 100 different economically significant uses derived from at least 39 different mangrove or mangrove-associated plant species, while Priyono et al. [82] provide many recipes for mangrove foods. Hossain [83] compiled a list of more than 39 mangrove and 11 mangrove-associated plants from the Bangladesh part of the Sundarbans mangrove forest with a description of various traditional uses of these plants. NTFPs from mangroves include many animal products, but animal products are generally fisheries-related (mangrove finfish, shrimp, crabs, bivalves and gastropods) and their economic significance is much better documented than plant-related NTFPs. Often, there are production data particularly for the most important finfish, shrimp, crabs and mollusks [36,84–86], even if they generally still tend to be poorly documented. In this paper, we focus on non-destructive plant-related products and uses.

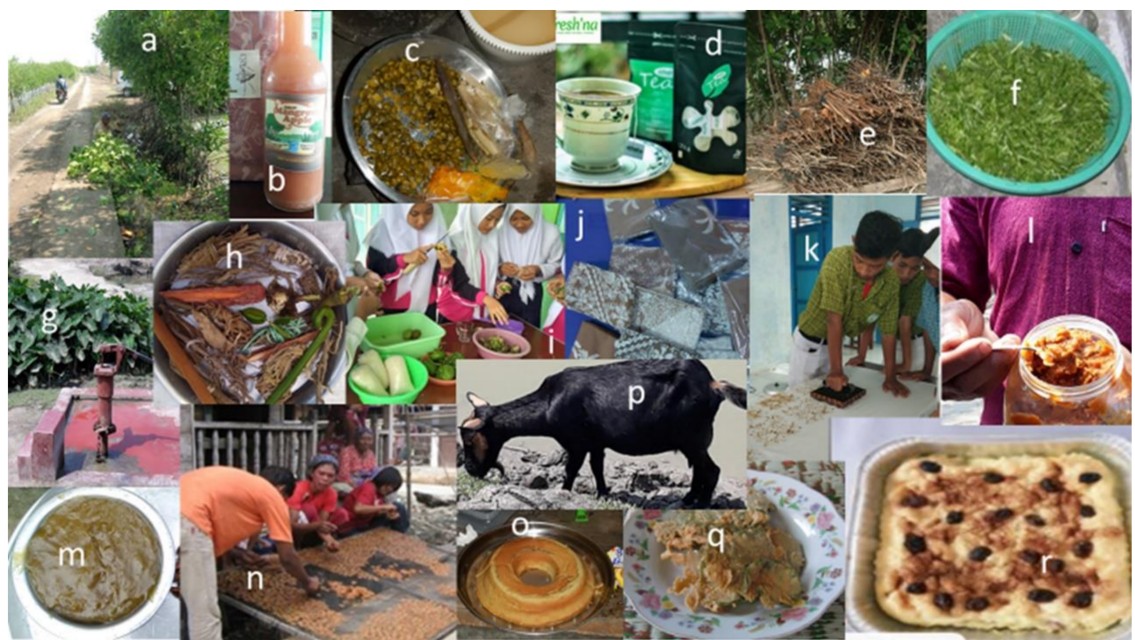

**Figure 2.** Examples of sustainable mangrove-based non-timber forest products (NTFPs), excluding fisheries: (**a**) *Rhizophora mucronata* fodder for livestock, Java, (**b**) bottled juice from *Sonneratia caseolaris*, Java, (**c**) food from *Avicennia marina* and *Bruguiera gymnorhiza*, (**d**) tea from *Acrostichum illicifolia*, Java, (**e**) sustainable firewood from *R. mucronata*, Java, (**f**) *A. illicifolia* leaf processing for tea or chips, (**g**) saline vegetable patch *Colocasia esculenta*, Bangladesh, (**h**) mix of mangrove medicinals including *R. mucronata* root, *Xylocarpus mekongensis* bark, *Vitex simplex* vine, *Thypha dominguensis* stalks and ear, Khulna, Bangladesh, (**i**) girls learning to make dessert from *S. caseolaris*, Java, (**j**) selection of handmade batik textile products, Java, (**k**) youths learning the batik techniques, (**l**) syrup from *Phoenix sylvestris*, Paikgacha, Bangladesh, (**m**) chutney from *Sonneratia apetala*, Khulna, Bangladesh, (**n**) *A. illicifolia* chips, Java, (**o**) *A. marina* cake, Java, (**p**) small saline-adapted Black Bengal goat, *Capra hircus*, (**q**) *Avicennia* leaf chips, Timbulsloko, Java, (**r**) *Nypa fruticosa* cake, Java. Photo credits: A.O.D.: a, e, g, j, l, p, q; R.F.: d, f, n, r; Kuswantoro: b, i, k; E.D.: c, o; M.U.A.: n, h.

Our expanded inventory provides a more comprehensive listing of sustainable non-wood uses for 203 Asia-Pacific mangrove forest plants (including mangroves and mangrove forest-associated plant species) (Supplementary Table S1). These amounted to 117 food-related uses; 33 livestock and fish feed-related uses; 126 domestic uses for crafts, as materials (including dyes and resins) or as ornamentals; and 133 health-related and/or medicinal uses; in total, 409 distinct uses of socioeconomic potential (Supplementary Table S1). This inventory is not intended to be exhaustive but does serve to show the enormous socioeconomically interesting potential, which has hardly been unlocked or developed, to provide coastal communities with sustainable mangrove-friendly livelihood alternatives. Of all these essentially sustainable possibilities, the only two that have so far been developed to commercially significant production volumes are the largely still unsustainable exploitation of both *Nypa* and honey [54,56,73].

## 4. Recent Trends in Mangrove NTFP Research and Development

The presumed value and potential of NTFPs having been established (above), the next question is whether this has translated into mangrove NTFP research and development interest. Figure 3 shows a bibliometric comparison between the annual production of mangrove research papers and all forest-related NTFP research listed in Scopus. The results show, based on a total listing of 20,742 mangrove publications and 2995 NTFP publications that have appeared since 1980, that mangrove-related research has consistently outstripped the interest in NTFP research for all forest

types combined. Furthermore, while the rate of production of mangrove-related research continues to increase, the rate of production of NTFP research appears to have flattened-off to a linear rate of increase (Figure 3).

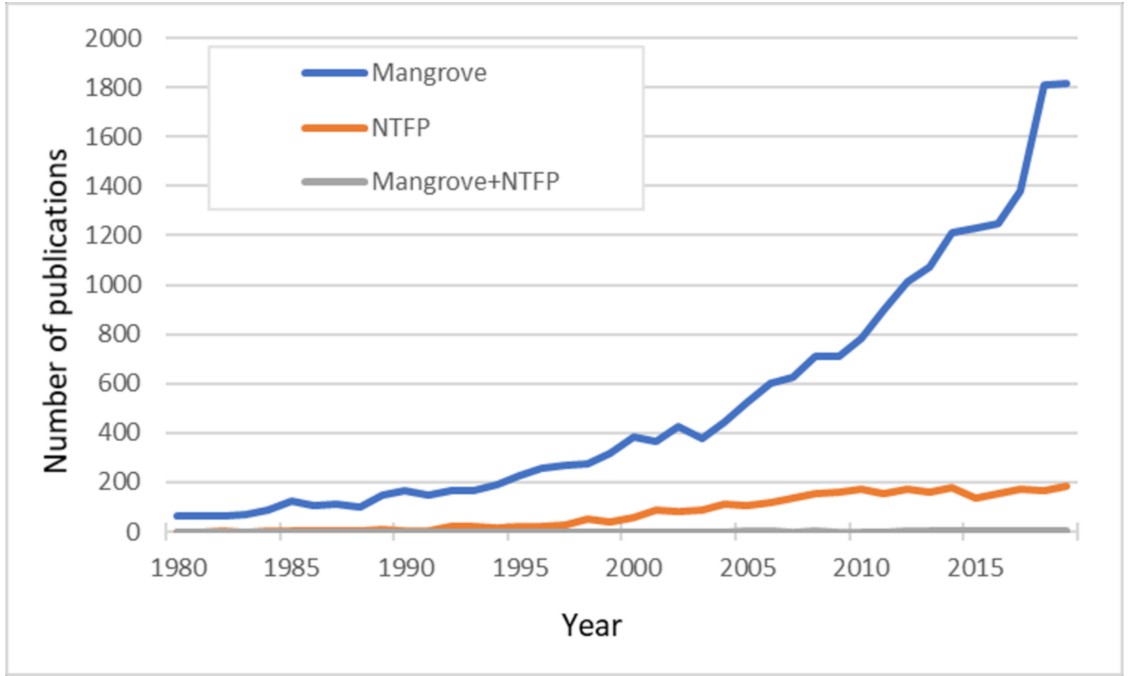

**Figure 3.** Trends in the annual appearance of publications in the total field of mangrove research and the total field of non-timber forest product (NTFP) research.

Figure 4 allows for a more detailed look at how a number of mangrove subdisciplines, including mangrove NTFP research, compare to all combined NTFP research. The number of Scopus papers listed per subdiscipline for the chosen period of time amounted to 20 mangrove + NTFP papers, 1211 mangrove + fisheries papers and 386 mangrove + restoration papers. The results show that since the 1980s, the annual output of publications dedicated to NTFP research has grown steadily to about 180 papers per year in 2018 and 2019. Quite understandably, considering the key fish nursery function of mangroves, mangrove fisheries research continues to be the most important livelihood-related area of mangrove research, amounting to over 130 publications annually at present (Figure 4). According to Dale et al. [87], research interest in the mangrove restoration/rehabilitation subdiscipline has increased rapidly since 2003. However, mangrove NTFP research has lagged well behind compared to (general) NTFP, mangrove fisheries and mangrove restoration work, and annual output has been less than one paper per year with no evident increase over time. The total number of papers jointly addressing mangroves and NTFPs remains very low. Our results confirm that while NTFP research (as applied to forest systems generally), mangrove fisheries and mangrove restoration research interest have grown steadily or even steeply in recent years, mangrove NTFP research has shown no significant development. Hence, our assessment provides quantitative corroboration of what (as pointed out above) other authors have indicated before.

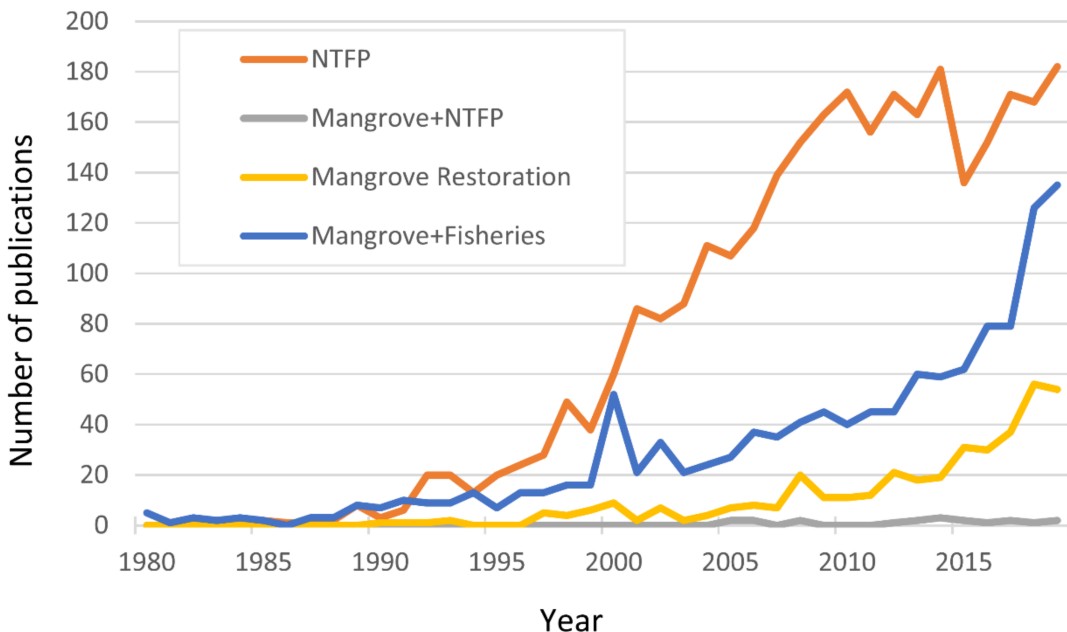

**Figure 4.** Trends in the annual appearance of publications in NTFP and key mangrove-related research areas (NTFP, mangrove + NTFP, mangrove + restoration and mangrove + fisheries).

## 5. Mangrove NTFP Characteristics and Consequences for Commercialization

### 5.1. Mangrove Ecological Factors

Mangrove NTFPs have various ecological characteristics that complicate and constrain their development. Firstly, being part of a diverse ecosystem, the multitude of products that mangroves provide are typically seasonal and are spread out at low density over large areas. One major consequence of this is that livelihood-focused approaches based on mangrove agroforestry have to be a small-scale harvest of a wide variety of specialty products. This again means that harvests likewise need to be widely spread, involving much manual effort and low-tech jobs [43]. Because of these characteristics, shared generally with other NTFPs, it is difficult to achieve economies of scale. Achieving scale effects is essential for these resources to be able to become more than subsistence resources and to elevate them above their status as "products of the poor" [57].

Secondly, local overexploitation is highly likely as plants and animals are thinly and patchily spread. This means that management to prevent local overexploitation is critical to sustainable livelihood development [88,89]. However, unlike terrestrial ones, mangrove forest ecosystems are fluid in a literal sense and extend their influence far beyond the forest tree strands. The logic follows, therefore, that mangrove forest resources require ecosystem-based holistic management systems and an adaptive management approach, which is often incompatible with inflexible legal frameworks. In addition, many mangrove ecosystems include endangered fauna that may be sensitive to disturbance. Hence, management systems need to include the use of protected areas, no-fishing areas and no-entry areas. An optimal choice of such areas needs to be based on detailed ecosystem assessments and inventories, which is a major challenge considering the data-poor setting typical to mangrove management situations. Even so, with proper management, mangrove NTFPs can be harvested with relatively limited impact on the forest [57].

Thirdly, due to their important bio-builder role, mangrove trees play a critical role in maintaining system integrity. Mangroves are forests that are found in a narrow zone at the land–sea interface. They export their main ecosystem benefits both seawards and landwards. Given such complexity, it is not surprising that they are undervalued especially when the development of their spatially explicit

value has remained largely neglected. Development of NTFP-based livelihood options has been so far neglected not only by the scientific community but also by funding agencies and the business sector.

Lastly, because of their bio-builder role, the use of production systems that involve removal of large swaths of mangroves (including integrally-associated habitats such as mudflats and creeks) for aquaculture, agriculture or other coastal developments needs to be limited in surface cover. Recent research suggests that mangrove aquaculture systems should be limited to less than 30% of total surface area to ensure sustainability [10]. In the Mekong Delta of Vietnam, restrained aquaculture development is permitted in protected mangrove areas, in which farmers are allowed to use 30–40% of the land for aquaculture (depending on the specific area) while 60–70% must be left as mangrove forest [10]. Given the reality of the high and mounting population pressure in the coastal zone of most tropical developing nations, the major challenge of this approach is to ensure integration of the 60–70% forested area into the economy. The development of mangrove NTFPs and their potential as high value-added end products could help to meet this challenge.

### 5.2. Mangrove NTFP Economic Factors

Market development for NTFPs has long been recognized as being particularly challenging as its success depends on many factors throughout the agricultural market chain. Kar and Jacobson [90], for instance, provide evidence from Bangladesh to illustrate how montane NTFP development is strongly limited by poor transportation, communications and financing. Even though *Nypa* harvest represents a \$16.25 million industry and 300,000 full or part-time jobs each year in Bangladesh, Islam et al. [73] highlight how poor infrastructure, dangerous collecting conditions, lack of fair market pricing and lack of capital greatly restrict benefits to *Nypa* leaf collectors of the Sundarbans mangrove forest. Studies from other forest settings likewise indicate that infrastructure, market access [91] and organization [92] are essential in order to generate good income from NTFPs. The question is how inclusive the value chain is and how equitably and rationally profits are being distributed.

NTFP value chains can involve many different actors and activities, all of which need to be efficiently linked for the marketing mechanism to function properly and allow effective commercialization of NTFPs [58,63,90]. Understanding the specific role of each actor is essential if conditions for the poorer actors in the chain are to be improved. There are at least four levels of intermediaries that complicate market access and limit profits for the poor collectors [57]. While negotiation skills are clearly critical to obtaining a good price [93], and NTFP collectors typically lack the market information necessary for successful price negotiation, the presence of more than one intermediary does not necessarily mean they only manipulate prices to their advantage. Many intermediaries provide important business services in remote settings, bear significant costs and expose themselves to great financial risks [94].

Hence, for the NTFP value chain to function well, successful fulfilment of supporting activities such as technological support, quality training, transportation, communications, credit access, market information and marketing support is essential [90]. Pandey et al. [57] have also addressed at length the key issues for NTFP value chain development including sustainability, monitoring, harvesting regime, post-harvest technology, value addition, diversity, market system, incentives and legal framework. One way of structuring the myriad of factors, actors and processes that are needed for successful NTFP development comes from the Food System Approach, a conceptual framework described by van Berkum et al. [95]. As highly complex agricultural production systems, food systems and NTFP systems share many characteristics. By using a systems approach, it becomes possible to analyze the relationships between, and effects on, different parts of the value chain system and also to incorporate the needs of all stakeholders through a participatory approach at all levels of the marketing chain [96]. It helps create an overview and clarify the transactions between all steps in the value chain between production and consumption and identifies both enabling and constraining factors at all steps in the value chain (Figure 5). Feedback loops can be identified as they are driven and affected by socioeconomic and environmental factors at different spatial and temporal scales at the different levels of the value chain. Such a systems approach further sheds light on non-linear processes in the food

system and possible trade-offs between policy objectives. Its ultimate purpose is to use the derived understanding to identify the different intervention options that can help overcome the identified constraints and make optimal use of any enablers.

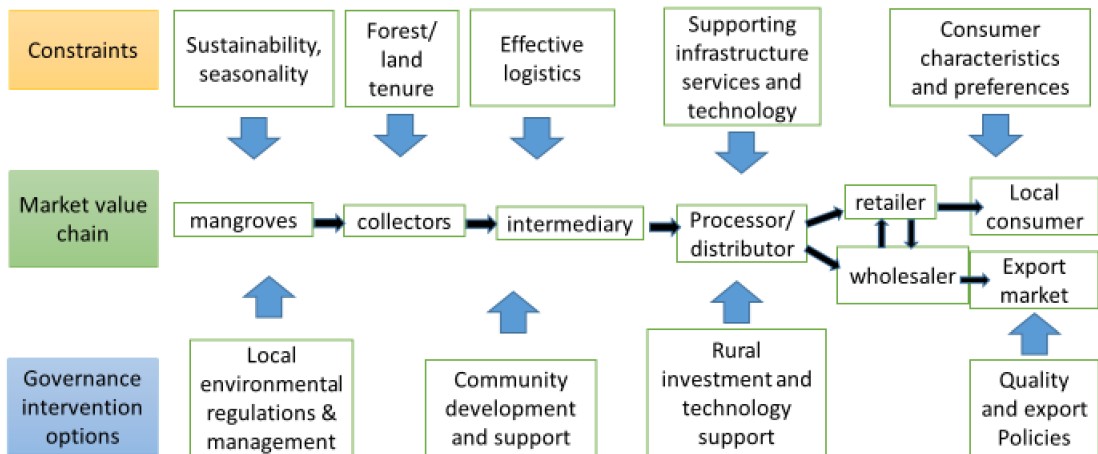

**Figure 5.** Mangrove value chain with the constraints and enablers that operate at different stages along the supply chain, modified from Pandey et al. [57] and Berkum et al. [95].

To successfully develop NTFPs, all stages in the system need to function properly and need to be developed gradually and in unison. In this, it is also essential to recognize the fact that mangrove community groups have long been marginalized [97–99] and that it is necessary to create a level playing field to ensure a participatory, inclusive and fair development. For many mangrove natural products to become more widely marketable and to go significantly beyond subsistence use, the development of intermediate-scale production technology for application at the village level is essential.

Even in cases where such technology can be made available, poor coastal communities will still need finances to be able to realize technological improvements. Finances can be obtained in many ways, but for cash-strapped communities, this will largely come down to subsidies from NGOs or some form of micro-credit arrangement, which today is widely provided by financial institutions [65]. In addition, group savings clubs are possible but are generally only suitable for more limited capital expenditures [100].

The most promising product lines for value-added local production are those that already have a wide support base either in terms of widespread familiarity and household use, partially established market chains and a widely available resource base, and for which readily available small-scale technology could help increase production scalability at the village level. The species of choice will differ depending on local environmental, socioeconomic and cultural factors that either enable or constrain sustainable commercialization. For instance, good candidates for development in the Sundarbans area of Bangladesh would seem to be mangrove honey and related products, *Nypa fruticosa* crafts and sweets, *Sonneratia apetala* chutney, *Sonneratia caseolaris* juice and *Phoenix sylvestris* syrup. For the north coast of Java, we suggest that some of the most interesting possibilities to start with might include the larger-scale production and marketing of a selection of mangrove teas, including *Acrostichum* tea, *S. caseolaris* juice, *Avicennia* leaf chips, and designer mangrove batik apparel. The importance of a participatory, inclusive and fair process for the success of such initiatives cannot be stressed enough [96,101–104]. Organizing production under village cooperatives should aid in achieving a minimum scale of production for a selection of different products, quality control, access to finances, branding and bargaining power.

Finally, a word of caution is in order because NTFP commercialization has often followed a "boom-bust" cycle. This has been found to be mainly caused by (a) inability of harvest rates to adapt quickly to product price changes, (b) overharvest, (c) domestication of the product and

(d) development of industrial substitutes [88]. Thanks to the recent trend towards sustainability and bio-economy, many NTFPs offer the possibility for high value-added products made within the communities living within forest areas [45]. By expecting and demanding sustainability and product authenticity, these trends offer a new perspective with which to address three of the four typical pitfalls of NTFP commercialization identified by Homma [88]. The dangers of the domestication of forest products into monoculture plantations to reduce production costs and increase production is a matter of particular concern. For instance, the successful commercialization of once "unique" forest products into widespread, mass-produced commodity products (e.g., oil palm, *Elaeis guineensis*, rubber, *Hevea brasiliensis*, chocolate, *Theobroma cacao*, and Brazil nut, *Bertholletia excelsa*) can hardly be argued to have brought much improvement to either sustainable forest management or marginalized forest peoples [88]. Thus, while we certainly do not present mangrove NTFP product development as a panacea to sustainable mangrove restoration, we do find that it has been disproportionally neglected and deserves more attention in light of the opportunities it offers for truly sustainable mangrove restoration and management.

## 6. Conclusions

Mangrove NTFPs have definite potential beyond their pure subsistence value and have an important role to play in reducing world poverty and in achieving sustainable development goals [96, 105]. However, for their potential to be realized, concerted efforts will be required towards scalable mangrove NTFP product development based on an integrated value chain approach, stemming from a systems perspective and based on a participatory, inclusive and fair product development process [96]. The whole product–market chain needs to be developed in unison for sustainable commercialization to be successful. The many pleas for attention to mangrove NTFP product development have remained largely unheard up to today. We here assemble evidence to show that recognizing, allowing and properly developing mangrove NTFP alternative livelihood options can contribute importantly to mangrove restoration success, and that they represent a still almost untapped opportunity to help bolster local support for mangrove restoration and conservation.

**Supplementary Materials:** The following are available online at http://www.mdpi.com/1999-4907/11/11/1224/s1.

**Author Contributions:** Conceptualization: A.O.D., A.V., S.W.K.v.d.B.; methodology: A.O.D., A.V.; formal analysis: A.V., M.P.; visualization: A.V., M.P.; data curation: C.J.K., M.N.I., M.I.A., M.N.A., M.U.A., S.R.H., R.F., Y.R.N., R.P., S.R. (Stijn Reinhard), E.D., E.K., S.R. (Sri Rejeki), C.T.v.S., C.V., M.P.; writing: A.O.D., A.V., M.P.; validation and review editing: S.W.K.v.d.B., C.J.K., M.N.I., M.I.A., M.N.A., M.U.A., S.R.H., R.F., Y.R.N., R.P., S.R. (Stijn Reinhard), E.D., E.K., S.R. (Sri Rejeki), C.T.v.S., C.V., M.P. All authors have read and agreed to the published version of the manuscript.

**Funding:** This work was made possible by core funding from grant KB-35-001-001 from the Wageningen University & Research "Food Security and Valuing Water" program, which is supported by The Netherlands' Ministry of Agriculture, Nature and Food Quality. Initial work was supported by The Netherlands Sustainable Water Fund, grant number: FDW14R14, as awarded to the Ecoshape Foundation, The Netherlands. Contributions by Wageningen Marine Research were additionally supported by supplementary funding from grant KB-36-003-012.

**Acknowledgments:** The authors thank Kuswantoro of Wetlands Indonesia for contributing two of the photographs and also wish to acknowledge two anonymous reviewers of an earlier draft for their critical and useful suggestions.

**Conflicts of Interest:** The authors declare no conflict of interest.

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
