# Peer review of "Non-Timber Forest Product Livelihood-Focused Interventions in Support of Mangrove Restoration: A Call to Action"

_forests, doi:10.3390/f11111224_

Round 1
Reviewer 1 Report
The article adresses a very important and up-to-date (and also sensitive) subject, the mangrove contributions to people. It defends a better valorization of non-timber forest products to the benefit of local people. This purpose is very relevant and in phase in particular with the priority work of the IPBES (on-going assessement of « Sustainable Use of Wild Species »)
Nevertheless, 3 points of the argument are debatable:
- The article is too much oriented on the example of Indonesia and more broadly on South-East Asia and Pacific.
- It is flawed in its argument about the systematic links between poverty and degradation.
- It does not enough consider the non economic (or instrumental) values of mangrove
The major hypothesis is not well argued
The fatal links between poverty and degragation have to be nuanced and adressed through the complex/multidimensional notion of poverty (no access to land, to credit, to market ? marginalisation and exclusion from the PA ?).
It contradicts itself by highlighting the major negative impact of shrimp farming ... and thus of capitalism and neo-liberal economy.
Another point is critical : the definition of mangrove as a forest. Rather it is a coastal weltand, with diverse habitats or components, from barren land (tanne) to open sea, with tidal channel (or bolon), mudflats, forest, etc.
The methodological and conceptual framework (notably Food System approcach) is relevant. But the manucript is lacking ethnography material to support its framework ; and the articles review, because of the biais, doesn’t compensate this weakness.
A key mangrove-based product is missing : salt (a « sustainable » product, easy to valorize and qualify)
More specific comments throughout the manuscript :
L 86-87 : Updated data regarding the loss of mangrove forest ?
50% of global coverage has already been lost : period ? time frame ?
35% having been lost worldwide during the 1980s and 1990s alone: what is the current global dynamic since 30 years ? Beyond Indonesia and South-East Asian countries, what is the status of mangrove forest in the last 20 years.
More and more publications underline the progression of the forest according to the scales and the context.
L 97 As shrimp pond construction is considered as the main driver of mangrove destruction, the authors have to nuance the links between poverty and overexploitation. The main actors of degradation are not the poor, but the rich, the private companies, neoliberalism and profit speculation.
L 109 restoration of livelihoods : YES a key stake and besides, the shift towards NBS (restoration of the infrastructure)
L112 economic integration : add socio-cultural values conservation
L 113-114 : this assertion is false or at least must be nuanced: « In much of the world’s tropics, overexploitation driven by poverty typically is the root cause of mangrove removal and degradation ».
L 118 may be unsustainable practices but also, more and more sustainable use/practices of mangroves at local or regional scales
L 149 : consumptive and non-consumptive output à I would say « extractive and non-extractive », because the mangrove-based products could be sold, as it is assumed trhoughout the manuscript
L 152 value of mangrove, beyond use or economic or instrumental value, relevance to consider the heritage and relational value
L 174-75 bibliograpy analysis à a lot of biais, that have to be highlighted
L249-250 I am not convinced that the “animal products are generally fisheries-related (mangrove finfish, shrimp, crabs, bivalves and gastropods) and are much better-known than plant-related NTFPs
L276 : « the only two that have yet been developed to commercially significant production volumes » à historically, it is the tannin, with long-distance commercialization along the East African Coast for the leather craftsmanship (cf. Rollet 1970 ; Cormier-Salem, 1999)
L365-373 : it a a good idea, but beyond the forest, the authors have not to take into account the mudflats, a key component for the mangrove restoration and a key habitat for a lot of species (avifauna and ichtyofauna). The hypersalted muldflats and barren lands, in hinterland, are also key source of a very valuable product, namely salt
L388 bio-economy : what does it mean ?
There are more and more incentives to qualify or give « signs of quality » for the mangrove-based products (scheme of labellisation, with traceability, charter of intrinsic and extrinsic quality, etc.), recognizing the local knowledge and know-how of women for extracting and processing the products.
A gender lens seems indispensable.
L448-471 : interesting consideration, illustrated with concrete cases, but I am skeptical about the criteria to choose the « good candidates » . « We » that means the researchers would be in charge of the selection ? There is an extensive litterature regarding the value-chain and the added-value schemes of « local products », underlining the importance of participatory (or inclusive and fair) process for the success of such initiatives

Reviewer 2 Report
General comments
I understand (and appreciate) that the aim of the paper is a double one: in the document a literature review and a call for action. A literature review should be based on traditional criteria of scientific paper writing, while a call for action can be based on rational hypothesis, not necessarily always supported by scientific results or by empirical evidence. However, the two components of the paper should be clearly defined as such in order to allow the reader to be able to recognize them. The call for action should be based on and follow the literature review while now the section 4.1 (Bibliometric assessment of NTFP research and development) is part of ch. 4 (Calls for NTFPs development) and separated from ch. 3 where the literature review is presented. Moreover, ch. 5 “Mangrove NTFPs characteristics and consequences for commercialization” comes after the Call for action and is a descriptive chapter on the same topics of ch. 3 with some overlapping and repetition.
Another serious problem on the structure of the paper is related to ch. 7 presenting a “Systems Approach to NTFP analysis and development”. Here, just close to the end of the paper, there is a presentation of the methodology proposed by Berkum et al., with reference to its implementation in the analysis of Bangladesh an Java NTFP markets (why only these two countries?). Presenting a method of value chain analysis after having described in two separate chapters of the paper (3 and 5) NTFP value chains seems to me unappropriated and very far from a coherent structure of a paper. An obvious question rise after reading this section: if the methodology proposed by Berkum et al. has a validity, why not to describe it in section 2 and applying it in describing mangrove NTFP? On the contrary, if it is only a potential future development of the studies in this sector, the methodology does not deserve a stand-alone chapter.
In section 3 (titled “Role of mangrove NTFPs and plurality of potential)” many arguments and citations are not specifically related to the mangrove environment; see, for example cit. no. 37, 38, 39, 41, 42, 43, ... This seems not consistent with the methodology as defined in the previous section 2: “The literature thus traced was reviewed to identify key sources on which to base our discussion of recent concepts applicable to the future of mangrove NTFP research and development”. I suggest to stick to this statement in presenting the literature review or to refer to a broader set of publications. In Google Scholar, for example, searching with “mangrove” and “non-timber forest products" it is possible to find (without records that are citations) 5,900 papers; with "mangrove” and “non-wood forest products" 2,900 papers.
Finally, there are two topics that I think deserve some attention: that one of domestication of mangrove species of economic potential (or semi-domestication with the related issue of how much the natural mangrove environment can be simplified in order to increase harvesting of some commercial species) and the topic of over-exploitation as a consequence of the intensification of use, a very well-known and much discussed issue by the literature on the development of NTFP-NWFP markets (Homma, 1992).
Comments and suggestions of minor importance
line 107: Is costal protection alternative to silviculture in mangrove rehabilitation? I think the two purposes are partly overlapping.
l. 145-…: it seems that fuel and charcoal are not considered NTFP while, by definition, they are. About definitions, I suggest to provide a clear definition of NTFP, also to clarify the role of fuel and charcoal.
l. 166: “…more recently…”. In Scopus the first paper on NWFP has been published in 1987; the first on NWFP in 1978, 9 years before. The first paper quoted in Scopus using the full concept of “non-timber (or -wood) forest product” is of 1967. In international literature (Scopus) NWFP is much more frequently used than NTFP.
l. 175: “… 2,995”. Making a check on Scopus (1st November, 2020) with “ntfp* OR non-timber AND forest AND product* OR nwfp* OR non-wood AND forest AND product*” 2,227 papers were found (2,058 in the period 1980-2019), not 2,995. This number should be checked, also in relation the contents reported in section 4.1.
l. 183-…: “The availability…”: statement already made; try to avoid repetitions.
l. 204: “costs”: better “payments”, “permits”, …. As mentioned in a following statement, there are always (at least informal, not explicit) costs.
l. 216: “In addition…”: in a literature review when statements like this one are presented they need to be supported with citations or [53] is supporting also to this statement?
l. 277: unsustainable exploitation conditions refer only to Nypa or also to honey?
l. 280: “…and services”: I suggest to write “…and other services” (mangrove NTFP are provisioning services)
l. 324: “20 papers…”: statement already made (l. 175); try to avoid repetitions.
l. 360-1: “they are undervalued especially when the development of their NTFPs value has remained largely neglected”: I think a bit too obvious statement.
l. 388: “...NTFPs more generally [37], offer ...”. Maybe better: “many NTFPs [37] offer ...”.
l. 390: “Mangrove-NTFPs pass along the value chain before reaching the end user”: obvious and very general statement. This part seems to me too general; concrete examples related to the mangrove context should be mentioned.
l. 396: “… unreliable supply…”: why there should be different conditions of supply if more added value is retained in the production area?
l. 399: “demand plasticity”: I don’t think this is rigorous and well-known concept in economics and social sciences.
l. 402: “... a well-managed resource”: as the authors already wrote, this (unfortunately) is not a specific pre-requisite for entering the international markets, esp. for exporting to emerging economies; there are hundreds of NTFP examples to support this statement.
l. 403: ”... can provide profit margins of 25% for each link”: the concept of “link” in a value chain is not rigorous (“transaction”?), so it is this percentage value that cannot be generalized. There are potential transactions along a value chain based on models from fully integrated to very fragmented, but there is not a direct and constant relations between number of transactions and “sustainable forest management enforcement” (by the way, another very vague concept).
l. 405: delete “that”.
l. 408: “NTFP value-chains involve ...”; better “NTFP value-chains may sometimes involve ...”.
Figure 5: “Effective producer organizations”: what’s the meaning of “producers”? If “producer organizations” refer to harvesters, the arrow that links them to “intermediaries” seems not logical; if “producer organizations” refer to the set of intermediaries, I wonder if we can use the concept of “organizations”: normally there is one or very few intermediaries and they are not organized, neither it seems logical to support formal organizations among intermediaries (their market power is already quite high).
l. 444-5: “.... links … need to be developed gradually and in unison“: this seems not fully consistent with what has been written previously (l. 392-3) where it was stressed the need for integrating and reinforcing local actors against external ones (assuming that they have more market power and they don’t need to develop “gradually and in unison“).
l. 450-1: “... efficient production process and an effective marketing channel for mangrove-NTFPs have yet to be developed”: quite obvious statement; it sounds quite repetitive.
Round 2
Reviewer 1 Report
The authors took my remarks into account and improved their article accordingly, with ad hoc clarifications and references.
I find their answer very well argued and I appreciated their openness and scientific rigor.
Reviewer 2 Report
I think the improvements and corrections made by the authors are adequate. The paper is suitable to a publication in Forests.